# Cancer Patients’ Survival According to Socioeconomic Environment in a High-Income Country with Universal Health Coverage

**DOI:** 10.3390/cancers14071620

**Published:** 2022-03-23

**Authors:** Vesna Zadnik, Tina Žagar, Sonja Tomšič, Ana Mihor, Katarina Lokar

**Affiliations:** Epidemiology and Cancer Registry, Institute of Oncology Ljubljana, 1000 Ljubljana, Slovenia; tzagar@onko-i.si (T.Ž.); stomsic@onko-i.si (S.T.); amihor@onko-i.si (A.M.); klokar@onko-i.si (K.L.)

**Keywords:** cancer, population-survival, socioeconomic deprivation, Slovenian Cancer Registry

## Abstract

**Simple Summary:**

The main aim of our paper was to quantify the potential impact of socioeconomic environment on cancer patients’ survival in Slovenia in the 21st century. Despite of universal health coverage and after accounting for basic demographic variables (age and gender), differences in stage at diagnosis, as well as the impact of the cancer treatment improvements over time, we show that cancer patients in Slovenia who have a lower socioeconomic status experience worse survival and increased mortality. The results of this analysis could help decision-makers to better understand inequalities in cancer burden and inform the development of policies to improve or resolve them.

**Abstract:**

Despite having an established systematic approach to population survival estimation in Slovenia, the influence of socioeconomic environment on cancer patients’ survival has not yet been evaluated. Thus, the main aim of our study was to quantify the potential impact of socioeconomic environment on cancer patients’ survival in our population in the 21st century. The net survival was calculated and stratified into quintiles of Slovenian version of the European Deprivation Index for all adult cancer patients diagnosed between 2004 and 2018 using the national cancer registry data. After accounting for basic demographic variables (age and gender), differences in stage at diagnosis, as well as the impact of the cancer treatment improvements over time, we found that cancer patients in Slovenia with lower socioeconomic status experience worse survival and have higher mortality. In particular, the odds of dying from oral, stomach, colorectal, liver, pancreatic, lung, breast, ovarian, corpus uteri, prostate, and bladder cancers, as well as for melanoma, leukemia, and non-Hodgkin lymphoma, are significantly higher in the socioeconomically most deprived group of patients compared to the most affluent group. The inequalities in cancer burden we found could help decision-makers to better understand the magnitude of this problem.

## 1. Introduction

The socioeconomic conditions people live in influence their health. When comparing groups with the lowest relative to those with the highest social standing, the former are at an increased risk of a multitude of diseases, and cancer is no exception [1]. There is ample evidence from across Europe revealing that cancer incidence in almost all common locations is associated with socioeconomic status (SES). Low SES is associated with increased risks of cancers of the lung, upper aerodigestive tract (oral, pharyngeal, laryngeal and esophageal), digestive tract (stomach, liver, gallbladder, and pancreas), as well as several genitourinary tract cancers (kidney, bladder, penis cancer in men and cervical cancer in women). On the other hand, breast, prostate, and skin cancers are more common in individuals of high SES. Smoking is the most important driver behind observed excess risks in low SES individuals and the magnitudes of the gaps are more often larger in men compared to women [2].

In addition to incidence, there is considerable variability in the survival of cancer patients among different cancer sites and populations [3]. The population-based survival of cancer patients is dependent on a multitude of factors, including those pertinent to the patient themselves (and their disease) as well as the performance of the healthcare system where the patient is treated. According to the researchers within the EUROCARE project, the European cooperative for studying population-based survival in cancer patients, important social determinants related to the healthcare system are healthcare workforce numbers along with diagnostic and treatment facilities, including mass screening programs [4]. When looking at this from a socioeconomic perspective, low education level and low social environment in a population represent an obstacle to timely diagnosis and accessibility of optimal treatment [1].

Epidemiological indicators consistently show that cancer is one of the most pressing public health problems in high-income countries, including Slovenia, a Central European country with two million inhabitants. Death from cancer is the number one cause of death in Slovenian men and second in women. In the past decade, on average, over 14,000 Slovenes developed cancer each year, and around 6000 died due to cancer. Currently, in Slovenia, there are over 120,000 prevalent cancer patients [5]. Given the risk of cancer increases substantially with age (two-thirds of patients are older than 65 when they are diagnosed with cancer), due to Slovenia being one of the fastest aging populations in Europe, the rising trend in cancer burden is expected to continue irrespective of potential changes in exposure to risk factors [5]. The association between SES and cancer incidence in Slovenia has already been documented: an increased incidence in socioeconomically deprived was found for head and neck, esophageal and lung cancers, while conversely, an increased incidence in the affluent group was seen for melanoma, breast cancer in women, and prostate cancer in men [6].

Slovenia operates a compulsory social health insurance system with universal population coverage, where all expenses for cancer diagnosis and treatment are fully covered. Along with the highly centralized cancer care facilities, including population-based cancer screening programs for breast (introduced in 2008 though only gradually expanding to finally cover the whole population in 2018), cervical (introduced fully in 2003) and colorectal (introduced fully in 2009) cancers, this should serve to provide equitable health care and consequently diminish differences in disease outcomes. The Slovenian Cancer Registry has been publishing in-depth data on the survival of cancer patients in Slovenia for more than 25 years. Monographic publications provide reviews of population survival for a specific period by individual cancer sites, accompanied by commentary from clinical experts who diagnose and treat those patients. Cancer survival has been improving steadily in Slovenia but is still lower than in some Western European countries [3,7]. Men, in particular, have seen some of the largest improvements in survival lately. The two most important determinants of cancer survival are age and stage at diagnosis; survival of patients with disseminated disease has not been improving in Slovenia [8]. Despite having an established systematic analytical approach to population survival estimation, in Slovenia the influence of SES on cancer survival rates is understudied.

Thus, we aimed to quantify the potential impact of socioeconomic environment on cancer patients’ survival in Slovenia in the 21st century. Considering Slovenia is a high-income European country with universal health coverage, no or comparatively small socioeconomic disparities were expected. The population-based data collected by the Slovenian Cancer Registry was used and the estimates could thus be controlled for basic demographic variables (age and gender), differences in stage at diagnosis, as well as the impact of the cancer treatment improvements over time.

## 2. Materials and Methods

The source of cancer patients’ data used in the analysis was the population-based Slovenian Cancer Registry (SCR). The SCR collects high-quality and complete data covering the whole population of Slovenia [9]. This is assured by linking the SCR to other high-quality population and health databases, including a daily linkage to the Central Population Registry for updating information on registered patients’ vital status and place of residence [9,10].

We analyzed survival of patients diagnosed between 2004 and 2018, stratified into 3 consecutive 5-year periods. Patients younger than 20 years or older than 95 were not included. The following data on all cancer cases meeting the inclusion criteria were extracted: sex, date of cancer diagnosis, age at diagnosis, location of cancer (following the International Classification of Diseases, 10th edition (ICD-10)), stage at diagnosis and vital status. Patients were followed either until the day they died, were lost to follow-up, or until the end of the study. The analyses were performed jointly for all cancer sites and further for distinct cancer entities—25 most common cancer sites defined by ICD-10 were considered. 

The association between the socioeconomic environment and cancer patients’ survival was evaluated using the Slovenian version of the European Deprivation Index (SI-EDI) for the year 2011 at the level of the National Assembly Polling Stations, the smallest administrative geographic units for which population data were available. A procedure suggested by Pornet et al. [11] was applied for SI-EDI calculation using the Slovenian version of the European Union Statistics on Income and Living Conditions (EU-SILC) survey [12] and the population census [13]. SI-EDI was categorized into quintiles (from most affluent in group 1 to most deprived in group 5) and assigned to cancer patients considering their address at diagnosis [6,14].

The variables described were extracted from the SCR database on 1 September 2021. Among a total of 201,739 cancer cases, there were 35,825 non-melanoma skin cancer cases. As is common practice in cancer survival analyses, these cases were excluded on account of being frequent but highly curable [3]. A further 1648 cancer cases were excluded due to the age limitations. Next, we excluded 707 cancer cases where the date of diagnosis was unknown and were registered in the SCR based on death certificate data only, and an additional 2215 cases where the date of diagnosis coincided with the date of death leading to zero survival time (mostly, in 91%, they were diagnosed at autopsy). Finally, 161,344 cancer cases were included in the analysis. It should be clarified that the number of cases does not represent the number of individuals, as one individual may have more than one primary cancer, defined according to European Network of Cancer Registries rules for cancer coding. The end of the study date was 31 August 2021 when the latest update on the vital status was performed in the Registry before data extraction. On this date, an individual could be either alive, dead, or lost from the vital statistics records. There were only 88 persons (0.05%) lost to follow-up after the diagnosis of cancer in the analyzed cohort, mostly due to emigration.

We calculated net survival according to Pohar–Perme. Net survival estimates how long after cancer diagnosis an individual would be alive if the only possible cause of death was cancer, i.e., cancer specific survival. This is achieved by inverse-probability weighting of person-time at risk, namely older individuals are assigned larger weights because they are more likely to die from other causes [15,16]. Because not every individual included in the analysis was followed-up for 5 years, the complete approach to survival estimation was used [17]. We reported 5-year net survival for the entire period as well as 10-year net survival for periods 2004–2008 and 2009–2013. Net survival is not presented for (sub)groups where there were less than 20 cases, and a number of cancer entities were excluded where most of the categories had less or around 20 cases. Analyses were performed with R software version 4.1.0 using the relsurv package version 2.2-6 [18]. The Cox proportional hazards regression model was applied for investigating the effect of socioeconomic deprivation on the survival of cancer patients [19]. In the multivariate Cox model, in addition to SI-EDI, the hazard ratios were adjusted for gender (males, females), period of diagnosis (2004–2008, 2009–2013, 2014–2018), age group at diagnosis (20–49, 50–74, 75–94) and stage at diagnosis (localized, regional, distant, or unknown extent of disease). In the analysis, the event was considered death by any cause. Cox modeling was performed using IBM SPSS Statistics Version 24 software. For calculating confidence intervals in the analyses, we used alpha 0.05.

## 3. Results

### 3.1. The Cohort

There were 161,344 cancer cases included in the analysis; 30% were diagnosed in the first 5-year period 2004–2008, 34% in the period 2009–2013, and 36% in the period 2014–2018. More than half of the cases were men (55%, 88,321 cases), the distribution by gender in the cohorts did not vary over time. Similarly, the distribution by stage and age groups in the cohorts was uniform in all analyzed periods (data not shown). In the case of solid tumors (C00–C80 according to ICD-10), most cases were detected in the localized stage (41%), 34% of cases were diagnosed in the regionally advanced stage, and 23% with distant disease. The remaining cases had an unknown stage. The percentage of unknown stages varied substantially by cancer site (from 0.05% for esophageal cancer to 9.2% for brain tumors) though it did not appear to be skewed in favor of cancer sites with either exceptionally good or bad prognoses. The majority of patients in our cohort were aged from 50 to 74 years at diagnosis (60%), only 11% were younger than 50. Out of the 25 cancer sites analyzed, the most numerous were prostate, breast, lung, and colorectal cancers.

The percentage of cases in each of the five national quintiles of deprivation was also uniform throughout the analyzed periods. The categories of most affluent, affluent, and medium SES each had 23% of cases, while in the deprived and most deprived groups, there were 19% and 12% of cases, respectively.

Table 1 shows the five-year net survival for 25 cancer sites and for all cancer sites combined stratified by five levels of SI-EDI for the most recent period. The corresponding results for the 2004–2008 and 2009–2013 periods are given in Appendix A. The 10-year net survival is also shown for the earlier two periods. Further, results of the same analyses stratified by gender are available in Appendix A. Comparing the corresponding confidence intervals in the above tables allows for the estimation of significance for the observed differences between SI-EDI categories. The estimations of the relative effect of socioeconomic deprivation on net survival are shown in Table 2, where the results are reported relative to the most affluent group and adjusted for gender, age group, stage, and calendar period.

### 3.2. All Cancer Sites Combined

The analysis for all cancers combined showed significant differences in net survival with an observed trend across categories of SI-EDI (Figure 1). The five-year net survival in the most deprived individuals diagnosed between 2014 and 2018 was 54.0% (52.5%−55.5%), while in the most affluent, it reached 63.2% (62.1%−64.3%), a difference of over 9 percentage points. However, the gap is narrowing; in the previous 5-year periods, the difference was even more prominent, 12 percentage points for 2004–2008 and 10 percentage points for 2009–2013. The odds for dying from cancer in the socioeconomically most deprived group of patients were 1.23 (1.21−1.26) compared to the most affluent group (Table 2). As expected based on previous analyses [8], the absolute survival was higher in recent years for each of the SI-EDI categories.

Figure 2 shows five-year net survival by SI-EDI categories stratified by gender, stage and age group at diagnosis. Women survived longer in all SES categories. Five-year net survival was 7 percentage points lower for the most deprived compared to the most affluent women. This difference was even larger in men—11 percentage points. The gaps between the five SI-EDI curves were even more pronounced for ten-year survival (Figure 1 and Appendix A). Survival of cancer patients diagnosed with distant metastases was low irrespective of SES, while there was a statistically significant difference between socioeconomically deprived and affluent patients diagnosed with localized (7 percentage points) or regionally advanced (9 percentage points) disease. There is no significant trend across SES categories in the survival of patients with unknown stages. A significant difference in survival between SI-EDI categories was also evident in all age groups—in the largest group of patients (aged 50 to 74 years at diagnosis), 66.8% (65.7–68.1%) of the most affluent patients survived five years, whereas survival in the most deprived group was only 57.2% (55.5–58.9%) (Figure 2).

### 3.3. Analysis by Cancer Sites

Considering specific cancers, a statistically different five-year net survival between the most affluent vs. most deprived groups with a negative survival trend from most affluent to most deprived was determined for colorectal and lung cancers in both genders combined for the last period analyzed (Table 1). Further, for the first period analyzed, in males, similar significant trends were observed for esophageal and prostate cancers (Appendix A). With respect to the hazard analysis, in Slovenia in the 21st century, the odds of dying from a specific cancer type in the socioeconomically most deprived vs. most affluent group of patients are significantly higher for oral, stomach, colorectal, liver, pancreatic, lung, breast, ovarian, corpus uteri, prostate and bladder cancers, melanoma, non-Hodgkin lymphoma, and leukemia. (Table 2).

For cancer sites with poor prognosis (lung, pancreas, gallbladder with bile ducts, liver esophagus, and brain tumors), where the five-year net survival in the most deprived group was below 20%, short-term survival could be more informative. Therefore, we have also calculated one-year survival for those sites (data not shown). We observed similar trends from the most affluent to most deprived in one-year survival. However, the differences were not statistically significant. The best one-year survival was seen for lung cancer at 49.0% for most affluent and 44.1% for most deprived. The worst one-year survival, on the other hand, was for pancreatic cancer with 28% for the two most affluent groups and 25.9% for the most deprived.

## 4. Discussion

### 4.1. Data and Methods

The Slovenian Cancer Registry is one of the longest continually operating cancer registries in Europe, with over 70 years of experience. It covers the whole population of Slovenia, with the registration of each new patient mandatory by law ever since its establishment in 1950. Consistently, quality indicators have shown an extremely high level of data accuracy and completeness, thus eliminating biases that might stem from underrepresentation of certain groups of patients [10]. 

The four most common cancers in the cohort (lung, colon and rectum, breast and prostate) represent almost half of all cancers in the Slovenian population, hence they had the greatest influence on the results of analyses examining all cancers combined. The structure of the analyzed cohort, considering cancer sites, age group, stage, and gender distribution is almost identical as reported in recent national and international comparisons of population survivals for Slovenian cancer patients [3,7,8]. Thus, the results of the present research can easily be extrapolated to previous studies we carried out.

For measuring SES, we used the Slovenian version of the European Deprivation Index—SI-EDI. EDI is an adaptable ecological deprivation index, first developed for France, and is based on an individual deprivation indicator as suggested by Townsend’s philosophy of relative deprivation [11,20]. To date, EDI has been adapted for Italy, Portugal, Spain, England, and Slovenia [21] and used in a number of studies on social inequalities in cancer [22,23,24], screening uptake [25], and health care access [26]. Launoy et al. recognized EDI in their recent monography as the fundamental tool for building evidence-based cancer policies in Europe [1]. Therefore, we consider the allocation of the cohort cases to SI-EDI categories was valid. One caveat with using SI-EDI is that the index has been developed for the year 2011, while our study period spanned from 2004 to 2018, though any bias that might be stemming from this is to some extent balanced by the year 2011 being right in the middle of the study period.

Regarding control of confounders in Cox modeling, unfortunately, aside from gender, age, stage, and period, we were not able to include data on patients’ comorbidities, which are known to be associated with lower survival [27] as well as SES [28] and could have in part influenced the observed results in survival disparities across SES categories. The observed gaps are, therefore, most likely a combination of factors associated with the patient (comorbidities, poorer treatment compliance, etc.) and the healthcare system (worse access to health services, biased treatment decisions, etc.) though we cannot speculate to what degree each played a part in specific cancer types.

### 4.2. All Cancers Combined

Socioeconomic disparities in the overall survival of cancer patients are not available in the Concord-3 global research [3]. On the other hand, the EUROCARE-5 study, including population-based data for cancer patients diagnosed between 2000 and 2007 from 29 European countries, showed that the five-year relative survival for all cancers combined for men and women was positively associated with macro-economic indicators, namely the Gross Domestic Product and Total National Expenditure on Health [7]. In Germany, researchers observed a difference in cancer mortality indicators between East and West Germany, which began disappearing after their reunion [29]. In an earlier ELDCARE study, survival of European cancer patients older than 65 was strongly positively associated with healthcare technology use, but not with healthcare workforce numbers, especially for cancers with a good prognosis. Furthermore, survival was better with a larger share of married older people (and worse with a larger share of widowed older people), meaning, at least in the elderly, social support has an effect on disease outcome [30]. The highest socioeconomic gradient is evident in rare cancers. These consist of different cancer entities with certain common issues regarding diagnosis and treatment, such as delays/uncertainty of diagnosis, few established lines of therapy, lacking expertise/dedicated facilities, as well as obstacles in research. Treatment for most rare cancers is not straightforward, requiring a multidisciplinary approach in dedicated expert centers. All these are less accessible for deprived populations [31].

### 4.3. Analysis by Cancer Sites

We found a negative survival trend from most affluent to most deprived for colorectal and lung cancers in both sexes, and prostate and esophageal cancers in men, whereas the Hazard analysis revealed a negative association with many more cancer entities (Table 1 and Table 2).

Systematic reviews published recently as part of a monography on social environment and cancer in Europe [1] provide a summary of evidence showing that for many cancer types, more deprived patients have lower survival. For colorectal cancers, a systematic European review found that patients with a lower SES experienced poorer survival [32]. In a recent study similar to ours, though focusing on digestive cancers in France, net survival was significantly lower for those residing in a more compared to the less deprived environment for cancers of the esophagus, liver, pancreas, and colorectal cancer, and additionally for stomach and bile duct cancers among females [33]. The excess mortality hazard ratio for colorectal cancer in men for less deprived vs. most deprived was estimated to be 1.2 in France and is very similar to what we have found in our study (Table 2: 1.25 for Slovenia). The findings from France are also in line with ours when it comes to esophageal cancers (hazard ratio 1.15).

For breast cancer, there is an overwhelming amount of data suggesting that a variety of known factors influencing survival (i.e., stage at diagnosis, treatment protocol, the status of hormonal receptors, lifestyle, comorbidities, and participation in screening) do not fully explain the socioeconomic gap [34]. In Slovenia, breast cancer patients from the most deprived category have odds of dying of 1.22 compared to the most affluent group. Similarly, for prostate cancer, the literature review points to a clear gap in survival for individuals with low compared to high SES [35]. Slovenian prostate cancer patients from the most deprived category have odds of dying of 1.27 compared to the most affluent group. Regarding lung cancer, a gradient of increasing survival with increasing SES has been observed in many European countries [36]. This is certainly true for Slovenia, where the most deprived patients have odds of dying of 1.14 compared to the most affluent group. SES inequalities in lung cancer, in particular, have been purported to stem from comorbidities due to smoking and other related lifestyle factors in addition to differential treatment choices [37]. Unfortunately, we were not able to control our results for smoking status or comorbidities and, therefore, cannot confirm whether this is the case also in Slovenia. Another group of cancers where survival is strongly associated with SES is head and neck cancer [38]. In Slovenia, the most deprived patients with oral cancers have odds of dying of 1.15 compared to the most affluent group, whereas for laryngeal cancer, the odds are as high as 1.25. Regarding non-Hodgkin lymphoma, we found increased odds (1.18) of dying for most deprived vs. most affluent, which is in line with findings from an English research group [39], which concluded that even after adjusting for comorbidities, socioeconomically deprived patients diagnosed with diffuse large B-cell and follicular lymphoma experienced worse survival.

Thyroid cancer was the only cancer entity where a reversed observation was found, i.e., the most deprived group of patients had better survival, though the results were not statistically significant. For the moment, we are unable to explain this finding and consider it merely coincidental.

## 5. Conclusions

Too many lives are cut short in Slovenia and other European countries on account of inequalities in cancer survival between population groups with different SES. Confronting social inequalities in cancer is thus one of the primary goals for the professional and lay communities as well as for decision-makers. Owing to the Slovenian Cancer Registry, which has been operating within the Institute of Oncology Ljubljana for over 70 years, Slovenia has an extremely long tradition of monitoring the burden of cancer and the quality of care for cancer patients. Drawing on the population-based Cancer Registry data, this article addresses the link between the socioeconomic environment and cancer in Slovenia. Considering that Slovenia is a high-income European country with universal health coverage, no significant socioeconomic disparities were expected. Despite universal healthcare system, after accounting for basic demographic variables (gender and age), differences in stage at diagnosis, as well as the impact of the cancer treatment improvements over time, we show that cancer patients in Slovenia who have a lower SES also experience poorer survival and increased mortality. We hope that our work will help national decision-makers as well as the broader public to better understand inequalities in cancer burden and inform the development of policies to improve or resolve them.

## Figures and Tables

**Figure 1 cancers-14-01620-f001:**
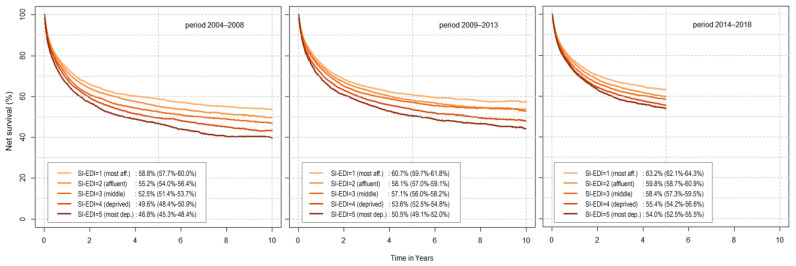
Net survival by Slovenian deprivation index (SI-EDI) with five-year survival and corresponding 95% confidence interval, by calendar period for both genders combined, Slovenia 2004−2018.

**Figure 2 cancers-14-01620-f002:**
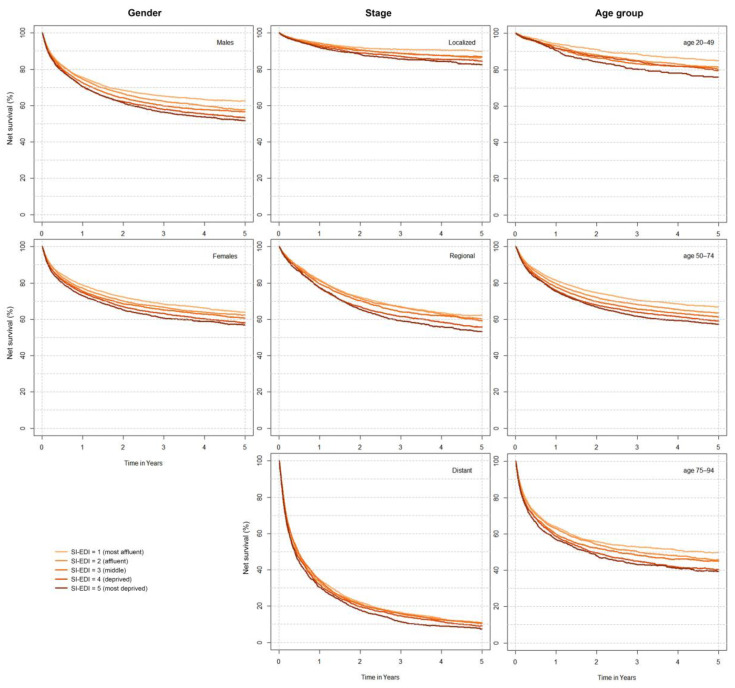
Net survival by Slovenian deprivation index (SI-EDI) and by gender, stage, and age group at diagnosis, Slovenia 2014−2018.

**Table 1 cancers-14-01620-t001:** Number of analyzed cancer cases (N) and five-year net survival with 95% confidence interval (CI) by cancer site according to ICD-10 classification and by Slovenian deprivation index (SI-EDI), both genders combined, Slovenia 2014−2018.

	SI-EDI = 1 (Most Affluent)		SI-EDI = 2 (Affluent)		SI-EDI = 3 (Middle)		SI-EDI = 4 (Deprived)		SI-EDI = 5 (Most Deprived)	
Cancer Site	N	Survival (95% CI)	N	Survival (95% CI)	N	Survival (95% CI)	N	Survival (95% CI)	N	Survival (95% CI)
*** All cancers (C00–C96 excl.C44)**	14,160	**63.2 (62.1–64.3)**	13,569	59.8 (58.7–60.9)	13,270	58.4 (57.3–59.5)	11,113	55.4 (54.2–56.6)	6773	**54.0 (52.5–55.5)**
Lip, oral cavity and pharynx (C00–C14)	336	51.1 (45.1–58.0)	333	53.8 (47.6–60.9)	420	55.2 (49.7–61.4)	410	44.7 (38.9–51.3)	262	40.9 (34.5–48.5)
Larynx (C32)	86	84.6 (71.2–100.5)	113	60.8 (50.8–72.6)	127	63.6 (53.9–75.1)	105	60.0 (50.0–72.0)	74	54.7 (41.8–71.5)
Esophagus (C15)	102	12.2 (6.9–21.4)	100	8.7 (4.3–17.7)	89	20.0 (12.3–32.7)	87	10.9 (6.0–20.1)	71	4.6 (1.5–13.8)
Stomach (C16)	475	33.3 (28.1–39.5)	463	33.5 (28.6–39.2)	489	28.2 (23.8–33.5)	453	33.1 (28.2–38.7)	299	22.1 (17.1–28.5)
*** Colon and rectum (C18–C20)**	1586	**67.7 (64.4–71.1)**	1608	63.7 (60.4–67.2)	1536	61.6 (58.2–65.1)	1258	60.3 (56.7–64.0)	824	**53.3 (49.0–57.9)**
Liver (C22.0)	154	14.4 (9.3–22.2)	172	11.5 (6.9–19.2)	172	6.4 (3.2–12.7)	147	13.7 (8.6–21.7)	105	12.2 (6.8–21.9)
Gallbladder and bile ducts (C23, C24)	239	13.9 (9.7–20.0)	240	14.9 (10.5–21.0)	212	17.1 (11.9–24.4)	167	10.5 (6.5–17.0)	102	15.2 (9.1–25.3)
Pancreas (C25)	499	7.5 (5.3–10.6)	458	9.6 (7.1–13.0)	473	5.1 (3.3–7.9)	372	6.5 (4.2–10.2)	182	6.9 (3.9–12.3)
*** Lung (C33, C34)**	1609	**22.7 (20.4–25.3)**	1582	22.0 (19.8–24.6)	1690	20.5 (18.3–22.8)	1444	21.1 (18.8–23.6)	897	**16.3 (13.7–19.4)**
Soft tissue (C38.0, C47–C49)	92	61.0 (49.9–74.6)	88	60.1 (48.6–74.1)	89	51.1 (40.1–65.0)	64	46.1 (33.4–63.8)	32	52.2 (36.4–74.8)
Cutaneous melanoma (C43)	807	93.8 (89.9–98.0)	710	90.0 (85.8–94.4)	591	92.8 (88.4–97.3)	465	85.9 (80.7–91.5)	259	86.8 (79.8–94.3)
Breast (C50) **	1655	88.6 (86.1–91.2)	1626	87.2 (84.5–89.9)	1651	86.3 (83.7–89.0)	1336	85.0 (82.0–88.1)	718	89.2 (85.1–93.5)
Cervix uteri (C53) **	112	71.7 (62.6–82.1)	122	64.4 (55.9–74.2)	114	75.8 (66.9–86.0)	119	65.0 (56.2–75.1)	82	77.0 (65.2–91.0)
Corpus uteri (C54)**	413	83.2 (77.9–88.8)	404	81.5 (76.4–86.9)	393	84.5 (79.3–90.1)	342	73.7 (68.1–79.9)	214	75.1 (67.0–84.2)
Ovary (C56) **	155	49.7 (41.5–59.5)	173	39.1 (31.5–48.4)	184	38.0 (30.8–46.9)	141	44.5 (36.2–54.6)	79	35.1 (24.8–49.6)
Prostate (C61) **	2097	98.3 (95.9–100.8)	1849	93.2 (90.5–95.9)	1791	94.8 (92.3–97.4)	1453	91.5 (88.6–94.6)	880	90.6 (86.8–94.5)
Testis (C62) **	141	99.0 (96.0–102.0)	129	94.3 (89.9–99.0)	126	96.5 (92.7–100.4)	98	96.6 (92.3–101.0)	64	99.4 (96.2–102.7)
Kidney (C64, C65)	434	65.3 (59.5–71.7)	432	69.5 (63.7–75.9)	421	63.0 (57.1–69.5)	374	66.7 (59.9–74.2)	246	61.9 (54.8–69.9)
Bladder (C67)	449	57.9 (51.9–64.6)	426	56.8 (50.0–64.6)	384	57.9 (50.8–66.0)	308	45.1 (37.9–53.6)	175	44.7 (35.7–56.1)
Brain (C70–C72)	198	13.5 (9.3–19.6)	190	10.7 (6.9–16.6)	179	20.2 (14.9–27.3)	158	16.3 (11.4–23.4)	94	15.0 (8.9–25.1)
Thyroid (C73)	252	95.4 (91.5–99.5)	222	94.6 (90.2–99.3)	218	93.1 (88.5–97.9)	164	94.9 (89.2–101.0)	112	97.6 (92.3–103.3)
Hodgkin’s lymphoma (C81)	56	80.3 (68.7–93.9)	70	82.4 (72.3–94.0)	55	79.1 (67.8–92.2)	38	81.6 (68.1–97.8)	16	/
Non-Hodgkin’s lymphoma (C82–C85)	537	66.1 (60.9–71.8)	493	64.6 (59.3–70.3)	452	62.4 (56.9–68.5)	371	59.2 (53.3–65.7)	230	62.9 (55.0–72.0)
Plasmacytoma (C90)	174	43.8 (35.3–54.4)	171	40.1 (30.9–52.0)	139	46.0 (36.9–57.4)	137	40.2 (30.4–53.3)	64	52.1 (39.3–69.1)
Leukaemia (C91–C95)	403	60.4 (53.9–67.6)	362	53.9 (48.0–60.6)	337	41.3 (34.8–49.0)	287	41.2 (34.1–49.8)	199	55.3 (46.6–65.7)

* A negative survival trend from most affluent to most deprived/5-year survival is statistically different between most affluent (SI-EDI = 1) and most deprived (SI-EDI = 5). ** Results are gender-specific (only breast cancers occurring in females are taken into account). N number of analyzed cancer cases; CI confidence interval.

**Table 2 cancers-14-01620-t002:** Hazard ratios with 95% confidence intervals by cancer site according to ICD-10 classification and by Slovenian deprivation index (SI-EDI), for both genders combined, Slovenia 2004−2018. Hazard ratios are adjusted for gender, age group, stage, and calendar period. The reference category is most affluent cancer patients (SI-EDI = 1).

Cancer Site	SI-EDI = 2 (Affluent)	SI-EDI = 3 (Middle)	SI-EDI = 4 (Deprived)	SI-EDI = 5 (Most Deprived)
All cancers	1.07 *	1.11 *	1.18 *	1.23 *
(C00−C96 excl.C44)	(1.05−1.09)	(1.09−1.13)	(1.16−1.20)	(1.21−1.26)
Lip, oral cavity	1.03	1.04	1.17 *	1.15 *
and pharynx (C00−C14)	(0.93−1.15)	(0.94−1.15)	(1.06−1.30)	(1.03−1.29)
Larynx (C32)	1.16	1.05	1.18	1.25
	(0.94−1.41)	(0.86−1.28)	(0.95−1.45)	(1.00−1.57)
Esophagus (C15)	1.12	0.82	1.01	1.15
	(0.94−1.34)	(0.69−0.98)	(0.85−1.20)	(0.95−1.39)
Stomach (C16)	1.04	1.03	1.05	1.13 *
	(0.96−1.13)	(0.95−1.12)	(0.97−1.14)	(1.03−1.23)
Colon and rectum	1.02	1.07 *	1.10 *	1.25 *
(C18−C20)	(0.97−1.07)	(1.02−1.13)	(1.04−1.15)	(1.18−1.33)
Liver (C22.0)	1.01	1.16 *	1.02	1.25 *
	(0.87−1.16)	(1.00−1.34)	(0.88−1.19)	(1.06−1.47)
Gallbladder and	0.95	0.96	1.09	1.14
bile ducts (C23, C24)	(0.85−1.08)	(0.85−1.08)	(0.97−1.24)	(0.98−1.32)
Pancreas (C25)	1.07	1.09 *	1.18 *	1.21 *
	(0.98−1.16)	(1.00−1.18)	(1.08−1.29)	(1.09−1.34)
Lung (C33, C34)	1.05 *	1.08 *	1.08 *	1.14 *
	(1.00−1.09)	(1.03−1.13)	(1.03−1.13)	(1.08−1.20)
Soft tissues	0.83	0.89	1.04	1.17
(C38.0, C47−C49)	(0.67−1.04)	(0.70−1.12)	(0.82−1.32)	(0.88−1.54)
Cutaneous melanoma	1.16 *	1.15 *	1.36 *	1.17 *
(C43)	(1.03−1.30)	(1.02−1.29)	(1.20−1.53)	(1.00−1.35)
Breast (C50) **	1.04	1.08 *	1.12 *	1.22 *
	(0.97−1.12)	(1.00−1.16)	(1.04−1.21)	(1.12−1.33)
Cervix uteri (C53) **	1.01	0.95	1.10	1.09
	(0.81−1.25)	(0.76−1.18)	(0.89−1.37)	(0.86−1.38)
Corpus uteri (C54) **	1.17 *	1.07	1.24 *	1.28 *
	(1.02−1.34)	(0.94−1.23)	(1.07−1.42)	(1.09−1.51)
Ovary (C56) **	1.12	1.13	1.23 *	1.28 *
	(0.97−1.29)	(0.98−1.31)	(1.06−1.43)	(1.08−1.52)
Prostate (C61) **	1.07 *	1.15 *	1.27 *	1.27 *
	(1.01−1.14)	(1.09−1.23)	(1.19−1.35)	(1.17−1.36)
Testis (C62) **	1.73	1.94 *	1.40	1.73
	(0.93−3.23)	(1.06−3.55)	(0.73−2.68)	(0.86−3.48)
Kidney (C64, C65)	1.11	1.23 *	1.29 *	1.14
	(0.99−1.23)	(1.11−1.38)	(1.15−1.44)	(1.00−1.30)
Bladder (C67)	0.98	1.08	1.15 *	1.21 *
	(0.89−1.08)	(0.98−1.20)	(1.03−1.27)	(1.07−1.37)
Brain (C70−C72)	1.08	0.95	1.02	1.03
	(0.95−1.24)	(0.83−1.08)	(0.88−1.17)	(0.87−1.22)
Thyroid (C73)	1.00	1.06	0.71	0.67
	(0.75−1.32)	(0.80−1.40)	(0.50−1.01)	(0.45−0.99)
Hodgkin’s lymphoma	0.87	1.12	1.07	1.57
(C81) ***	(0.56−1.34)	(0.72−1.75)	(0.66−1.72)	(0.90−2.73)
Non-Hodgkin’s lymphoma	1.00	1.05	1.18 *	1.18 *
(C82−C85) ***	(0.90−1.11)	(0.94−1.17)	(1.05−1.32)	(1.03−1.36)
Plasmacytoma	1.09	1.04	1.02	1.15
(C90) ***	(0.94−1.26)	(0.90−1.21)	(0.87−1.19)	(0.96−1.39)
Leukaemia	1.11	1.31 *	1.36 *	1.31 *
(C91−C95) ***	(1.00−1.24)	(1.17−1.46)	(1.22−1.52)	(1.15−1.49)

* statistically significantly different from 1. ** not adjusted by gender. *** not adjusted by stage.

## Data Availability

The data presented in this study are available in this article (and Appendix A).

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
