# Peer review of "Cancer Patients’ Survival According to Socioeconomic Environment in a High-Income Country with Universal Health Coverage"

_cancers, 2022, doi:10.3390/cancers14071620_

Round 1

Reviewer 1 Report

I congratulate the authors on the manuscript, focused on a very interesting topic: socioeconomical inequalities in cancer survival. Please find below some comments and suggestions to improve the article.

Major comments:

  • According to lines 156-159, 10% of the cases had missing stage at diagnosis. This can be an important limitation. Are stages missing because the prognosis is poor? Or maybe the stage is missing for cancers with high survival (e.g. melanoma)? I suggest doing a short descriptive analysis and/or computing survival for the cases with missing stage (e.g. adding a new graph in Figure 2 for missing stage).
  • In some cancer sites with poor prognosis (e.g. lung, pancreas), short-term mortality and survival are very useful. I suggest computing or discussing short-term indicators for survival (1-3 years) to assess the impact of socioeconomic inequalities in those anatomical sites.

Minor comments:

Please be aware that pages are not correctly numbered.

  • Introduction
    • In lines 33-34, the sentence “…the former at an increased risk…” should be “…the former ARE at an increased risk…”
    • I suggest to better redact the sentence “… prevalence of cancer patients in Slovenia is over 120,000” (lines 59-60), reflecting that 120,000 is the number of prevalent cancer patients.
    • It would help the reader if you could add the year the screening programmes for breast, cervical and colorectal started.
  • Methods
    • It is not clear to me why the acronym SI-EDI was chosen. Where does “SI” comes from? Maybe SL-EDI is more appropriate.
    • In lines 113-114 it is not clear if the five groups created for categorizing SI-EDI are quintiles, please make it clear. I suggest also changing it in the abstract (line 19).
    • In lines 141-142 please add the version of R and the version of the package {relsurv}.
    • I would suggest replacing “age” with “age group” in line 146. Also in the title of Figure 2 and line 200, to improve the interpretation of the figure.
  • Results
    • Please consider adding “Number of cases” at the beginning of the title of Table 1. Also, it is important to mention the confidence intervals 95 % in the title (not only in the footnote).
    • Breast appears as a cancer site in Table 1, where both genders are combined. Please clarify if you are including cases of breast cancer in both sexes, or only in women. Also, please note that breast cancer does not appear in Supplementary Table 1 (which is also for both sexes combined).
    • Reading the footnote in table 1 and Supplementary Tables 1-3, I can assume you are computing the p-value of trend for each anatomical site. I suggest adding the p-value as a new column in those tables.
    • “Lungs” appears in table 2, and Supplementary Tables, while “Lung” appears in Table 1. Please use the same name for that category across all tables.
    • The title of table 2 should include the word “mortality”.
    • In figure 1, the title and the numbers shown are referred to 5-year net survival, but the figure shows the net survival from 0 to 10 years. I suggest removing “Five-year” from the title, and specify that the numbers shown inside each graph are referred to 5-year. Add “Net” to the title of the Y-axis. And the legend can improve changing “EDI” by “SI-EDI”. These comments can also be useful to improve Figure 2.
    • Figure 2. Please add the variables “Gender – Stage – Age group” as a header for each column of graphs, to improve readability.
  • Discussion
    • The exhaustivity of the data from a population-based cancer registry like the Slovenian Cancer Registry is certainly one of the main strengths of the study and can be included in the discussion.
    • While the use of SI-EDI is completely justified, the authors need to acknowledge as a limitation the fact that the index is referred to the year 2011, while the cancer data covers previous years.
    • In the section “4.3. Analysis by cancer sites”, the (statistically significant) finding of lower mortality among the most deprived group is not discussed.
    • For the discussion of the results found for lung cancer (lines 289-291) I would add a sentence or two about the lower survival for most deprived patients, that might be explained due to the presence of comorbidities or difficulties in access to treatment (see references: Finke et al. Socioeconomic differences and lung cancer Survival—Systematic review and meta-analysis. Front. Oncol. 2018; Afshar et al. Factors Explaining Socio-Economic Inequalities in Cancer Survival: A Systematic Review. Cancer Control 2021; Redondo et al. Socio-Economic Inequalities in Lung Cancer Outcomes: An Overview of Systematic Reviews. Cancers, 2022).
  • Supplementary Tables:
    • The title of Supplementary Table 2 reads “… by pe-riod …”.
    • In some numbers there are points instead of commas as thousand separators: e.g. in Supplementary Table 3, number of cases for all cancers in 2014-2018 (“6.245” instead of “6,245”) and breast cancer (“1.643” instead of “1,643”).
    • The order of cancer sites is not the same as the used in Tables 1 and 2.

Reviewer 2 Report

It was a pleasure to read such a well-written paper with robust scientific investigation on such an important topic. There are only a few comments to be made.

  • line 57-59 there is no reference for these statistics
  • line 158 the percentages don’t add up, does that mean some of the solid tumours were unclassified?
  • line 311 should read Despite universal, or in spite of universal
  • Other studies into the socioeconomic associations with cancer also investigate co-morbidities (IHD, chronic kidney disease, obesity, diabetes), as they often result in a greater burden of disease in low SES and are known to decrease survival for patients with cancer. Was this data captured in the databases that you used, and should this be discussed?

Reviewer 3 Report

Greetings:

Thank you for your study. I found the reporting of the data very detailed especially for the different types of cancers and cancer categories. I was wondering if there was a more streamlined approach to showing the findings so the reader doesn't have to continue to refer back to a previous paragraph/section to see differences or similarities. 

Author Response

This manuscript is a resubmission of an earlier submission. The following is a list of the peer review reports and author responses from that submission.